# Overcoming the water oxidative limit for ultra-high-workfunction hole-doped polymers

Qi-Mian Koh[1,3], Cindy Guanyu Tang[2,3], Mervin Chun-Yi Ang [1], Kim-Kian Choo[1], Qiu-Jing Seah[1], Rui-Qi Png [2✉], Lay-Lay Chua [1,2✉] & Peter K. H. Ho [2]

It is widely thought that the water-oxidation reaction limits the maximum work function to about 5.25 eV for hole-doped semiconductors exposed to the ambient, constrained by the oxidation potential of air-saturated water. Here, we show that polymer organic semiconductors, when hole-doped, can show work functions up to 5.9 eV, and yet remain stable in the ambient. We further show that de-doping of the polymer is not determined by the oxidation of bulk water, as previously thought, due to its general absence, but by the counter-balancing anion and its ubiquitously hydrated complexes. The effective donor levels of these species, representing the edge of the 'chemical' density of states, can be depressed to about 6.0 eV below vacuum level. This can be achieved by raising the oxidation potential for hydronium generation, using large super-acid anions that are themselves also stable against oxidation. In this way, we demonstrate that poly(fluorene-*alt*-triarylamine) derivatives with tethered perfluoroalkyl-sulfonylimidosulfonyl anions can provide ambient solution-processability directly in the ultrahigh-workfunction hole-doped state to give films with good thermal stability. These results lay the path for design of soft materials for battery, bio-electronic and thermoelectric applications.

[1] Department of Chemistry, National University of Singapore, Singapore, Singapore. [2] Department of Physics, National University of Singapore, Singapore, Singapore. [3]These authors contributed equally: Qi-Mian Koh, Cindy Guanyu Tang. ✉email: ruiqi@nus.edu.sg; chmcll@nus.edu.sg

Recently, there has been much interest in sorbed water as a possible trap of hole carriers in polymer organic semiconductors[1–3]. Water is typically present at the level of $10^{19}$ cm$^{-3}$ in these materials, as small hydrogen-bonded clusters, when exposed to the ambient[4]. This concentration increases greatly in bulk-doped materials, where charge-counterbalancing anions must also be present. An emerging domain of such materials is the self-compensated, charge-doped polymers[5,6]. They can provide chemically 'tuneable' work functions in fine steps across an unprecedentedly ultra-wide range through a mixture of semiconductor core and ion effects[7–9]. The hole-doped polymer members exhibit hydration levels of a few $H_2O$ molecules per hole (i.e., $10^{20}$–$10^{21}$ cm$^{-3}$), considerably more than even ionic liquids. The relevant question is then not about the possible degradation of carrier mobility but the irreversible trapping of carriers into 'chemical traps'. Different from physical traps, these transform the carrier into a new chemical species, resulting in irreversible de-doping of the semiconductor. In this report, we examine the consequences of such traps on the ultimate stability of ultra-high workfunction (UHWF), hole-doped states of organic semiconductors.

Pioneering analysis by de Leeuw et al.[10] has yielded an important insight for the general stability of p- and n-doped organic semiconductors. The thermodynamic stability of hole-doped states may be limited by the potential of the water-oxidation reaction (WOR):

$$1/2\,H_2O_{(\ell)} \rightarrow 1/4\,O_{2(g)} + H^+_{(aq)} + e^-. \qquad (1)$$

For air-saturated, pH-neutral liquid water, the reaction has an electrode potential of 0.815 V vs. the standard hydrogen electrode[11]. This corresponds to 5.25 eV below vacuum, setting the Fermi level (FL) of the $H_2O/O_2$ couple. It is much shallower than the adiabatic valence-band edge of water, which lies at 6.95 eV (Supplementary Fig. 1). When the vacuum work function $\phi$ of the hole-doped semiconductor exceeds 5.25 eV, hole transfer to water becomes thermodynamically favourable, generating hydronium ions that deplete hole density in the surface region of the semiconductor. Thus, we have denoted $\phi > 5.25$ eV as 'UHWF', in view of this vulnerability.

However, bulk water is generally not present in organic semiconductors, neither on their surface nor in their interior. The water present exists primarily as small hydrogen-bonded clusters associated with the hydrophilic ion clusters inside the polymer. Thus, the relevant quantity that determines whether these water clusters act as chemical hole traps is their ionization energies ($I_E$), not FL of bulk water. Although we frame our analysis for self-compensated, hole-doped polymer organic semiconductors, the results are generally applicable to ion-containing soft materials.

Empirical observations have established that UHWF states of polymer semiconductors can be stabilized by certain non-nucleophilic counter-anions[5]. This was attributed to the low hygroscopicity, i.e., the ability to sorb water molecules, of non-nucleophilic anions. However, subsequent work reveals that hygroscopicity remains high enough to de-dope the semiconductor several times over[12,13]. Yet, this does not occur. We show here the reason is the large downshift of the donor levels of the water molecules associated with these hydrated anions, blocking their action as chemical hole traps. UHWF states can coexist with the water that is inevitably present, even under glovebox conditions, whereas hygroscopicity remains generally a destabilizing factor. Consequently, UHWF organic semiconductor films that are both ambient processable and thermally stable can be realized, for work functions as large as 5.9 eV, well beyond the limit imposed by bulk water. In contrast, UHWF metal oxide films, such as $MoO_3$, degrade rapidly in the ambient

to give a final $\phi$ of ca. 5.3 eV or so[14–16]. Finally, we also show that the UHWF polymer films exhibit a surprising tolerance even for liquid water, through the formation of reversible, surface dipole that protects against hole de-doping.

## Results and discussion

**Putative chemical hole traps.** Ion-containing polymers are well known to exhibit a dichotomous morphology comprising hydrophilic ion clusters dispersed in a hydrophobic polymer matrix[17]. As a subset, self-compensated, hole-doped polymers have their holes charge-counterbalanced by adjacent ion clusters with an overall negative charge. The inevitable hydration in air, and even in the glovebox, leads to association of molecular water with the anions and cations, but primarily at the surface of the ion clusters[13]. In the usual case where the 'spectator' cations are small univalent ones, such as {Li$^+$, Na$^+$, K$^+$}, the water is primarily hydrogen-bonded to anions residing at the surface[13]. These hydrated anions can then be idealized as X$^-$ (H$_2$O)$_p$, for mono-valent anions, where $p$ is the hydration number. Similarly, the hydrated mono-cations can be idealized as M$^+$ (H$_2$O)$_p$. However, these are of little relevance even if they were to occur, because the strong cationic potential greatly downshifts, i.e., stabilizes, donor level of the associated water outside of the practical UHWF range. Likewise, the H$_2$O molecules that bridge between anions and cations, i.e., X$^-$(H$_2$O) M$^+$, also do not limit the oxidative stability.

Thus, we can identify three primary chemical hole traps and four possible trap sites as follows:

(i) isolated water clusters, hole-trapping reaction represented by Eq. (2);
(ii) anion itself, whether dry ($p = 0$) or hydrated ($p \geq 1$), Eq. (3);
(iii) water cluster associated with anion, Eqs. (4a) and (4b), depending on hole-trapped product; and
(iv) covalent acid, which may occur as impurity, by-product of doping or of de-doping, Eq. (5):

$$(H_2O)_{q(s)} \rightarrow H^+(H_2O)_{q-1}HO^\bullet_{(s)} + e^-; \qquad (2)$$

$$X^-(H_2O)_{p(s)} \rightarrow X^\bullet(H_2O)_{p(s)} + e^-; \qquad (3)$$

$$X^-(H_2O)_{p(s)} \rightarrow X^-H^+(H_2O)_{p-1}HO^\bullet_{(s)} + e^-; \qquad (4a)$$

$$X^-(H_2O)_{p(s)} \rightarrow X{-}H(H_2O)_{p-1}HO^\bullet_{(s)} + e^-; \qquad (4b)$$

$$X{-}H(H_2O)_{p(s)} \rightarrow X^\bullet H^+(H_2O)_{p(s)} + e^-; \qquad (5)$$

The product of Eq. (2) is hydronium ion, denoted for simplicity in the rest of this report as 'H$_3$O$^+$', and hydroxyl radical HO$^\bullet$. The product of Eq. (3) is a neutral radical of the anion; Eqs. (4a) and (4b) hydronium ion and protonated anion (i.e., covalent acid), respectively, together with hydroxyl; and Eq. (5) hydronium and a radical of the anion.

**Condition for stability.** To evaluate whether a putative species acts as a chemical hole trap, we consider its solid-state adiabatic electron-detachment Gibbs free energy $\Delta G_{D,s}$ within the matrix of the hole-doped semiconductor (Fig. 1a). If $\Delta G_{D,s} < \phi$, hole transfer to the trap becomes thermodynamically favourable, both quantities referenced to the external vacuum level (VL). The FL of holes lies $\phi$ below VL. The surface dipole is small—typically <0.1 eV for self-compensated, hole-doped polymers[8,9]—thus, no correction of $\Delta G_{D,s}$ is required. The trap species varies explicitly with $p$, the number of water molecules associated with the anion under evaluation. Further, its energetics depends strongly on the number of spectator ion pairs $r$, i.e., (M$^+$X$^-$)$_r$, embedded in the

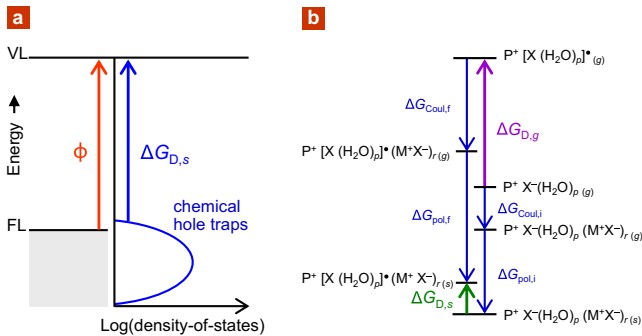

**Fig. 1 Chemical density of states (DOS) and computational cycle. a** Schematic of chemical DOS model, given by the distribution of $\Delta G_{D,s}$ for the chemical hole traps, which may also evolve with time. Irreversible hole trapping occurs when $\Delta G_{D,s}$ over a signification portion of the DOS becomes smaller than $\phi$, corresponding to emergence of that trap over the observation time. **b** Computational Hess cycle for $\Delta G_{D,s}$, given by: $\Delta G_{D,g} + \Delta(\Delta G_{Coul}) + \Delta(\Delta G_{pol})$, where $\Delta(\Delta G_{Coul}) = \Delta G_{Coul,f} - \Delta G_{Coul,i}$ is the differential Coulomb stabilization by local ion cluster, and $\Delta(\Delta G_{pol}) = \Delta G_{pol,f} - \Delta G_{pol,i}$ is the differential polarization stabilization by matrix, including carrier screening. $P^+$ represents polymer polaron; $[X (H_2O)_p]^\bullet \in \{[X^- H^+(H_2O)_{p-1} HO^\bullet], [X - H (H_2O)_{p-1} HO^\bullet], [X^\bullet (H_2O)_p]\}$.

local ion cluster hosting the trap species. For each $(p, r)$ pair, $\Delta G_{D,s}$ also depends on trap configuration, i.e., arrangement of the ion and water constituents, which can further dynamically change with time. Thus, $\Delta G_{D,s}$ occurs as a distribution, weighted by the likelihood of various configurations, which we call a 'chemical' density of states (DOS) (Supplementary Note 1). For the purpose of stability evaluation, however, only the frontier portion of this chemical DOS that is accessible over the observation time scale $\tau$ is required. The relevant $\tau$ is 10 h for processing stability, but longer for shelf stability. Whether the hole-doped state is stable or not depends on whether this portion lies deeper or shallower than FL.

**Computation of $\Delta G_{D,s}$.** The water–ion cluster configuration space is gargantuan. To make computation tractable, we assume separability and additivity of the dominant interactions as follows. (i) Quantum mechanics interactions, including hydrogen bonding, are of very short range. Geometry and energetics of the putative chemical trap, including its gas-phase adiabatic electron-detachment energy $\Delta G_{D,g}$, were computed for 'molecular' entities in the gas phase by density functional theory (DFT)/CAM-B3LYP/6-31++G($d,p$), neglecting possible distortions in the solid state. (ii) Multipole Coulomb interactions with ions in the local ion cluster are of medium range and are treated as 'background' Coulomb potential at the trap site. Ion-cluster configurations were sampled by an inexpensive hybrid MM2 molecular mechanics/semi-empirical PM3 methodology to obtain this correction term at the PM3 level: $\Delta G_{Coul,f} - \Delta G_{Coul,i}$. (iii) Matrix polarization, including charge carrier screening, is of long range. This was treated in the classical polarizable continuum model in the generalized Born approximation to obtain the final correction term: $\Delta G_{pol,f} - \Delta G_{pol,i}$. Thus, $\Delta G_{D,s}$ can be split into two parts by the Hess cycle (Fig. 1b): gas-phase $\Delta G_{D,g}$ energetics from (i), combined with corrections from (ii) and (iii). See also 'Methods' section, 'Computational methodologies'.

The selected DFT hybrid functional/basis set has been validated for electron-detachment energies of several hydrated anion complexes[6], hydrogen-bonding geometries and energies (Supplementary Tables 1-1 and 1-2), and appearance energies of water clusters (Supplementary Table 2). We investigated a model anion set, $X^- \in \{PF_6^-, Tf_2N^-, MsTfN^-, CF_3SO_3^-, MeSO_3^-, Me$

$(MeO)PO_2^-, CF_3CO_2^-, MeCO_2^-\}$, spanning super-acid to weak-acid anions, where Tf is $CF_3SO_2$, Ms is $CH_3SO_2$ and Me is $CH_3$. With the exception of $PF_6^-$, tethering with alkylene or alkyleneoxy chains to give self-compensated, hole-doped polymers would not significantly alter $\Delta G_{D,s}$. The computed $\Delta G_{D,g}$ values for $0 \leq p \leq 5$ are given in Supplementary Table 3. The anions are embedded in model ion clusters $P^+ X^-(H_2O)_p (Na^+X^-)_r$, where $P^+$ denotes the positively charged polymer backbone fragment, and $Na^+X^-$ is the spectator ion pair, for $0 \leq r \leq 3$. The differential Coulomb corrections are given in Supplementary Table 4. The corrections for differential matrix polarization, including carrier screening, are outlined in Supplementary Note 2. These amount to a stabilization of the final state by ca. 0.2 eV, allowing for the possibility that the hole comes from outside the locality of the chemical trap. The associated entropy effects would contribute an additional $-nk_BT$, where $n$ is of order unity. This amounts of ca. 0.1 eV for de-doping of 10%, smaller than the estimated overall uncertainty in our results of ±0.2 eV.

**Effective donor energies.** The smallest absolute value of $-\Delta G_{D,s}$ found for each putative chemical hole trap at specified $(p, r)$ gives its effective donor energy, which marks the edge of its chemical DOS. This corresponds to the most potent hole-trapping configuration for each composition. The results are plotted in Fig. 2 as vertical colour bands for $r$, separated into discrete bars for $p$. The computed results for $(H_2O)_q$ clusters dispersed in the same organic matrix are also shown. These exhibit $\Delta G_{D,s} \gtrsim 7.5$ eV (see also Supplementary Fig. 2). Thus, isolated $(H_2O)_q$ clusters per se do not limit the stability of practical UHWF states. Equation (2) is ruled out. In contrast, the hydrated anions exhibit a wide range of donor energies. The shallowest donors can certainly give irreversible hole trapping. The highest-occupied-molecular-orbital wavefunction, together with the empty counterpart of the singly-occupied-molecular-orbital (SOMO*) wavefunction of the corresponding hole state $[P^+ X^-(H_2O)_p]^{\bullet+}$, is shown in Supplementary Figs. 3 and 4, respectively. The predicted trend for $\{PF_6^-, Tf_2N^-, CF_3SO_3^-\}$ at $r = p = 0$ correlates well with their oxidation potential trend measured by linear sweep voltammetry[18]. This supports our simple model. Detailed analysis reveals several key trends for the following distinctive classes.

*Trends in super-acid to strong-acid anions: $PF_6^-$, $Tf_2N^-$, $MsTfN^-$, $CF_3SO_3^-$, $CH_3SO_3^-$.* For $p = 0$, hole trapping occurs on the anion; for $p = 1$, on the hybridized water…anion complex; and for $p \gtrsim 3$, on the associated water cluster. As these anions have little propensity to covalently bind protons, even at large $p$, the WOR produces an anion…hydronium pair, exemplifying Eq. (4a). The donor level upshifts strongly from the corresponding isolated water clusters. It also upshifts strongly with $r$, due to Coulomb interactions with the ion cluster stabilizing hydronium generation. Thus, water is a 'killer' of UHWF states, because of its association with anions. $CF_3SO_3^-$ shows an anomaly, because the hole is predicted to trap on the anion even at $p = 3$.

*Trends in weak-acid anions: $Me(MeO)PO_2^-$, $CF_3CO_2^-$, $CF_3CO_2^-$.* These anions have a strong tendency to bind protons, so the charge-separated anion…hydronium pair cannot exist. Hole is predicted to always trap on the anion, whether hydrated or not, exemplifying Eq. (3). Hence, the donor level is determined largely by the anion. In contrast to the super-acid to strong-acid anions, the donor level downshifts strongly, i.e., stabilizes, by hydration, due to charge delocalization and also spectator-ion stabilization effects. Coulomb interactions with the ion cluster always destabilize neutralization of the anion, whether by electron detachment or protonation. Although perfluoroalkyl carboxylates appear

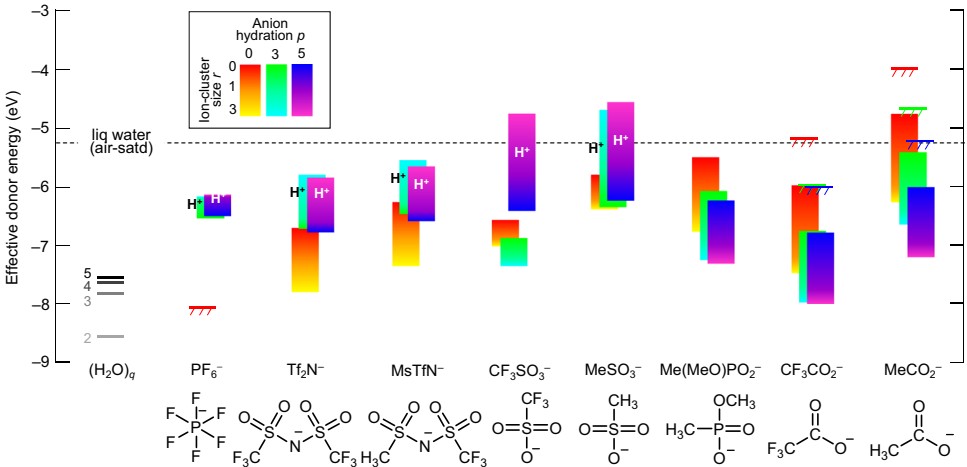

**Fig. 2 Effective donor energies of putative chemical hole traps.** Donor energy is plotted as the smallest absolute value of $-\Delta G_{D,s}$ for each of eight trap species at specified $(p, r)$, in vertical colour bands for $r$, separated into discrete bars for $p$. Energy is referenced to vacuum level in the standard semiconductor picture. Estimated uncertainty ($\sigma$), ±0.2 eV. Model: $P^+ X^-(H_2O)_p (Na^+X^-)_r$ clusters embedded in a conducting organic matrix. Results for $(H_2O)_q$ clusters embedded in the same matrix are shown as reference. Matrix parameters: dielectric constant, 3.0; carrier screening length, 12 Å. Hole-trapped product legend: $H^+$, hydronium; unlabelled, neutral species of radical or protonated anion. Kinetic donor band edges are represented by coloured levels with hashed lines, upshifted by irreversibility—anion fragmentation—estimated for a wait time of $10^5$ s. Horizontal dashed line marks the oxidative stability limit of air-saturated liquid water (5.25 eV). The thermodynamic donor band edge for $PF_6^-$ lies deeper than −9 eV.

capable of stabilizing UHWF states, decarboxylation of their carboxyl radicals would upshift the kinetic donor level quite significantly[6]. This eliminates the use of the carboxylate anions in UHWF materials.

*Trends in protonated anions.* The donor level of the protonated anion for $X^- \in \{Tf_2N^-, MsTfN^-, CF_3SO_3^-, CH_3SO_3^-\}$ lies at ca. 5.9 eV, for $p = 3$, exemplifying Eq. (5). This is almost invariant with X due to the similarity of X–H bond strengths. Thus, protonated anions set an ultimate limit to UHWF stability if they are present in the system. For the weak-acid anions, this reaction leads to identical outcome as Eq. (3).

**Anion design rules.** Thus, the potency of the WOR to de-dope an UHWF state is predicted to strongly depend on the anion, its hydration, and the local ion-cluster size. The ultimate ambient stability limit may lie at 6.1 eV with hydrated $PF_6^-$. For practical tethered anions, this may be lowered to 5.9 eV, as exemplified by $MsTfN^-$ for $r \lesssim 2$, broadly independent of $p$. These values are considerably beyond the assumed 5.2 eV limit for bulk water. On the other hand, the popular sulfonate is a distinctly poor choice, whether perfluoroalkyl or not, imparting an oxidative stability limit that is poorer than bulk water. Carboxylate anions may also be unsuitable, because of low inherent oxidative stability or kinetic upshift due to decarboxylation. Therefore, we recommend the following anion design rules for UHWF states: (i) employ large, non-nucleophilic, super-acid anions that have a high inherent oxidative potential and (ii) limit their organization with spectator cations, if any, to only small ion clusters by restricting their mobility.

**Evidence for WOR: tethered sulfonate.** We confirm here for sulfonate-compensated UHWF states that the WOR is indeed the stability-limiting reaction, both in air and in the glovebox. Poly (9,9'-bis(3'-sulfonatopropyl)fluorene-2,7-diyl-1,4-phenylene-N-(m-trifluoro-methylphenyl)-amino-1,4-phenylene) was employed as model (mTFF-$SO_3$; chemical structure in Supplementary Fig. 5). This is a member of the important triarylamine–*alt*–fluorene copolymer family that exhibits wide $\phi$-tuneability and simple spectroscopic properties suited for fundamental

studies[19]. An mTFF-$SO_3$ film was hole-doped by contact with an anhydrous acetonitrile solution of nitrosonium hexa-fluoroantimonate (NOSbF$_6$), washed with anhydrous acetonitrile, and spin-dried to give the self-compensated state with doping level (DL) of 0.85 hole per repeat unit ($h^+$/r.u.) and an initial $\phi$ of 5.4 eV. The infrared spectra collected at each step are shown in Fig. 3a. The difference spectra are also shown, shaded with components. The polaron infrared-active-vibration (IRAV) bands at 1560 and 1160 cm$^{-1}$ are characteristic of the hole-doped polymer. During doping, $NO^+$ exchanges for $Na^+$ and then hydrolyses to sulfonic acid. As a result, the –$SO_3^-$ modes disappear, replaced by –$SO_2OH$ modes, whose wavenumbers are well known[20]. The 2000–3300 cm$^{-1}$ spectral region contains the stretching modes of hydronium ions ($\nu$ '$H_3O^+$') of various forms—including Eigen and Zundel[21–24]—and also those of $H_2O$ molecules ($\nu$ $H_2O^*$) strongly hydrogen-bonded to the sulfonic acid groups[25]. These assignments have been authenticated by references (Supplementary Figs. 6 and 7). The hydrogen-bonded sulfonate–sulfonic acid–water behaves as a hydrated sulfonic acid network (spectra i, ii and iii), apparently due to proton transfer on the vibrational time scale. This is schematically illustrated by state 2 in Fig. 3b. Proton substitution has been confirmed by x-ray photoelectron spectroscopy (XPS) (Supplementary Fig. 8)[26].

After 16 h in a nitrogen glovebox ($pH_2O < 1$ p.p.m.), the film sorbs sufficient moisture to yield a water band (3200–3650 cm$^{-1}$; spectrum iv), which corresponds to state 3. The value of $p$ is ca. 1–2[13], which appears large enough for adjacent water clusters to overlap and support proton dissociation[27]. The $\nu$ '$H_3O^+$' band and the 960–1120 cm$^{-1}$ band intensify. The latter is characteristic of '$H_3O^+$' Zundel proton shuttle and Eigen umbrella modes[21–24]. The value of $r$ is estimated to be 1. Thus, the donor level is at ca. 5.4 eV and the film lies at the edge of stability. It shows de-doping at the rate of 0.1 $h^+$/r.u. over this time, based on optical spectroscopy (see 'Methods' section, 'DL evaluation') and bleaching of IRAV band intensities. Evidently, this UHWF state de-dopes slowly even inside the glovebox, generating hydronium.

Exposing this film to humid nitrogen (65% relative humidity (RH), 22 °C) increases hydration to 3–4 $H_2O$ per sulfonate (spectrum v)[13]. De-doping rises to 0.2 $h^+$/r.u. in 1 h, building up '$H_3O^+$' to give state 4. At longer times, the sulfonate–sulfonic

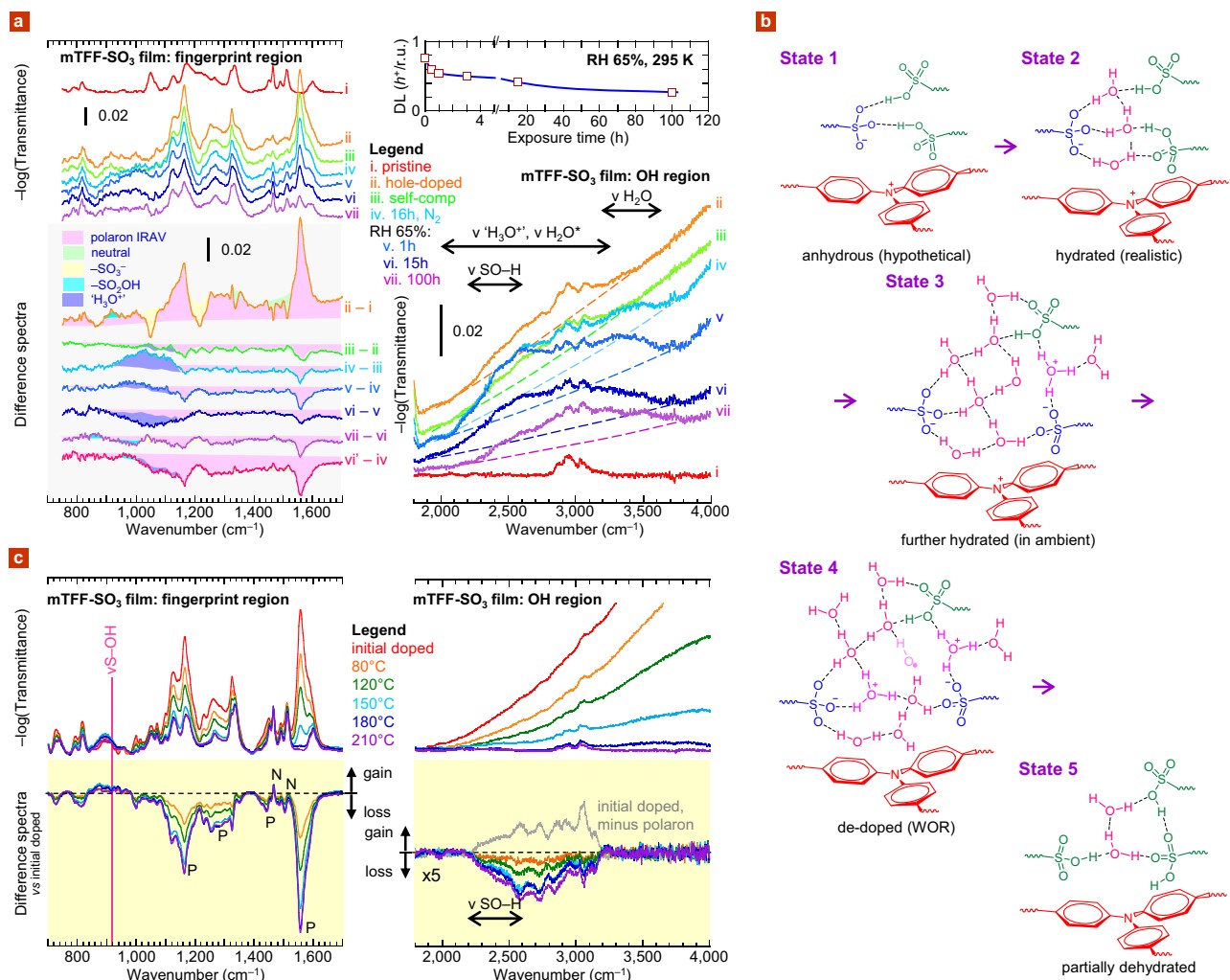

**Fig. 3 Evidence for ubiquitous water-oxidation reaction in sulfonate-compensated polymer films. a** Glovebox/ambient de-doping of mTFF-SO$_3$ film. Transmission FTIR spectra in dry nitrogen (<1 p.p.m. H$_2$O; i–iv) then wet nitrogen (65% RH, 295 K; v–vii), together with difference spectra (grey background). Spectrum (ii) was collected after contact doping with NOSbF$_6$ in anhydrous acetonitrile; (iii) doping level 0.85 $h^+$/r.u.; and (vi') after keeping film overnight in glovebox, following (vi). OH region: dashed line denotes polaron absorption; 'H$_3$O$^+$', hydronium ion in various forms; H$_2$O*, water in strong hydrogen bond–donor environment. Film thickness, 0.36 µm; substrate, Si wafer. **b** Schematic of local ion cluster: state **1**, anhydrous (hypothetical); **2**, hydrated in glovebox (realistic); **3**, further hydrated in ambient; **4**, de-doped by water-oxidation reaction; **5**, partially dehydrated after proton–anion recombination. **c** Thermal de-doping of mTFF-SO$_3$ film in glovebox. Transmission FTIR spectra after 10 min bake at each temperature step, together with difference spectra (yellow background). Initial film was obtained by contact doping with N($p$-BrC$_6$H$_4$)$_3$SbCl$_6$ in anhydrous acetonitrile, DL ≈ 0.7 $h^+$/r.u. Band labels: P, polaron IRAV; N, neutral. OH region: polaron absorption subtracted to visualize OH changes. Film thickness, 8 µm.

acid–water network appears to undergo reorganization by hydronium recombination with sulfonate to give sulfonic acid, lowering overall hygroscopicity (spectrum vi cf. v). We returned the film to the glovebox and acquired its spectrum under dry conditions (spectrum vi'). The difference spectrum (vi' − iv) yields the negative polaron IRAV spectrum overlaid with negative 'H$_3$O$^+$' features and positive $\nu$ S-OH band (variable, in the vicinity of 915 cm$^{-1}$). No change occurs in the vulnerable $p$-position of the pendant ring (1450–1530 cm$^{-1}$). After 100 h in humid nitrogen, DL decays to 0.25 $h^+$/r.u. (spectrum vii) and $\phi$ falls to 5.2 eV. The ion cluster now comprises a mixture of states 4 and 5. Similar results are obtained for poly(9,9'-bis(3'-sulfonatopropyl)fluorene-2,7-diyl-1,4-phenylene-$N$-($sec$-butylphenyl)-amino-1,4-phenylene), where $meta$-CF$_3$ is replaced by $para$-sec-C$_4$H$_9$ substitution. This polymer appears even more vulnerable to de-doping. Even though initial $\phi$ is 5.15 eV (DL, 0.8 $h^+$/r.u.), it decays to 5.0 eV (0.4 $h^+$/r.u.) after a few hours in the ambient. These results confirm that WOR is the key destabilizing mechanism for sulfonate-compensated hole-doped states, both in the ambient and in the glovebox.

**Tethered sulfonate: thermal de-doping.** Baking a hole-doped mTFF-SO$_3$ film in the glovebox also causes its de-doping via the WOR. This occurs because of the water present at equilibrium inside film at room temperature, even in the glovebox. A film of doped mTFF-SO$_3$ was heated to increasing temperatures on a digital hotplate in the glovebox and its infrared spectra collected ex situ (Fig. 3c). The initial spectrum shows the IRAV (1100−1600 cm$^{-1}$) and $\nu$ H$_2$O* (2200–3200 cm$^{-1}$) bands as before. Molecular water ($\nu$ H$_2$O) remains absent throughout. Baking induces loss of H$_2$O*, but also de-doping. This generates sulfonic acid, as evidenced by the intensification of $\nu$ S-OH and $\nu$ SO-H bands. Thus, the film de-dopes from state 2 to state 5 with increasing temperature, following Eq. (4b), likely driven by entropy. No other chemistry occurs. Complete recovery of the de-

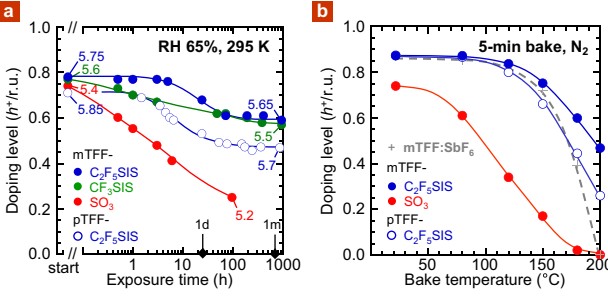

**Fig. 4 Doping stability characteristics of UHWF $R_F$SIS-compensated polymer films. a** Doping-level stability with log(exposure time) in ambient air. Start and end points are labelled with work function. **b** Doping-level stability with bake temperature in glovebox. Dashed line gives characteristic for mTFF:SbF₆ film as reference. Film thickness, 50 nm. Doping level was evaluated from neutral $\pi \rightarrow \pi^\star$ and polaron intensities, with an estimated uncertainty of ±0.05 $h^+$/r.u.

doped film to its initial undoped state occurs upon alkali treatment (spectrum iii–ii, cf. i–ii in Supplementary Fig. 6). Thus, de-doping can occur till completion in the glovebox through the WOR, even for a low degree of hydration of ca. 1 $H_2O$ per sulfonate.

**Tethered $R_F$SIS: enhanced stabilities.** Guided by the design rules, we design two perfluoroalkyl-sulfonylimidosulfonyl ($R_F$SIS) derivatives—CF₃SIS and C₂F₅SIS—as anions attached to the end of a $C_3$-tether. Both have practically identical $\Delta G_{D,g}$ slightly smaller than $Tf_2N^-$ (Supplementary Table 3; S/N 3 and 5), but C₂F₅SIS has more restricted mobility. We attached them to poly (triarylamine-*alt*-fluorene) semiconductor cores with *meta*-CF₃ (mTFF) and *para*-CF₃ (pTFF) substitutions (Supplementary Fig. 5) to achieve the desired extreme UHWF[19]. Different from sulfonates, $R_F$SIS enable quantitative hole doping in solution without the side reaction of proton substitution (Supplementary Fig. 9). C₂F₅SIS-compensated hole-doped polymers can be isolated dry and re-dissolved in a variety of solvents, including acetonitrile and propylene carbonate.

The $R_F$SIS anions indeed impart better ambient stability to the UHWF state. Although DL of the mTFF-SO₃ film decays from ca. 0.8 to 0.5 $h^+$/r.u. in 3 h (65% RH, 295 K), the mTFF-C₂F₅SIS film decays little over this period, despite its much larger $\phi$ of 5.75 eV (Fig. 4a). The mTFF-SO₃ film continually decays to 0.25 $h^+$/r.u. in 100 h, similar to that in wet nitrogen. On the other hand, the mTFF-C₂F₅SIS film shows a step decay with a characteristic time of ca. 25 h, levelling off at 0.6 $h^+$/r.u. after 75 h. The CF₃SIS analogue appears marginally less stable with an earlier decay onset, but DL still remains 0.6 $h^+$/r.u. at 1000 h. A hole-doped pTFF-C₂F₅SIS film with an initial $\phi$ of 5.85 eV also remains unchanged in air for 2 h, highly remarkable for such an UHWF. It eventually decays to 0.45 $h^+$/r.u. with a characteristic time of ca. 15 h, levelling off at a final $\phi$ of 5.7 eV. The greatly improved stability also manifests in the glovebox (Supplementary Fig. 10). These results are unprecedented.

The successful suppression of the short-time decay opens a window for ambient processing of UHWF hole-doped states. These states can persist for extended periods in ambient air—a few hours for $\phi$ up to 5.85 eV and over 1000 h for $\phi$ up to 5.7 eV. The step-like decay that occurs on the characteristic time of ca. 20 h for tethered C₂F₅SIS suggests a slow relaxation of its hydrated ion clusters, perhaps reorganization to larger clusters (i.e., larger $r$), upshifting the chemical DOS for hydronium generation. Hydration itself is fast and completes on the time scale of minutes (Supplementary Fig. 11).

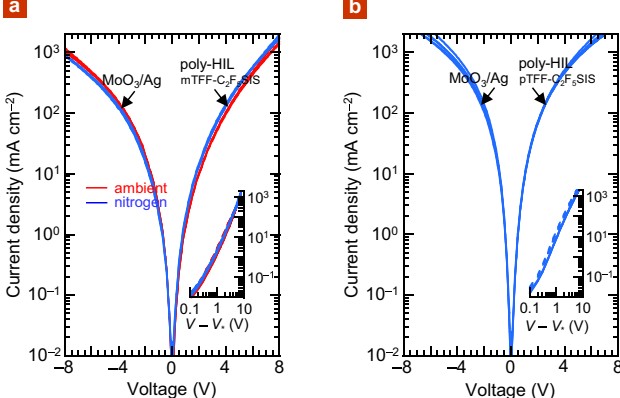

**Fig. 5 Hole injection characteristics of UHWF $R_F$SIS-compensated polymer films.** Device structure: glass/ITO/25 nm poly-HIL/110 nm PFOP/ 2 nm MoO₃/Ag, comparing hole injection from MoO₃ (negative bias) and from the polymer hole injection layer (positive bias) for: **a** mTFF-C₂F₅SIS as poly-HIL, spin-cast in air ('ambient') or glovebox ('nitrogen') and **b** pTFF-C₂F₅SIS as HIL, spin-cast in glovebox. Bias voltage is applied to poly-HIL, hole injecting contact is labelled for each branch. Representative lo-to-hi second sweep data (5 V s⁻¹) are shown for three devices each. Inset shows log($J$/mA cm⁻²) vs. log($V − V^\star$) plot, where $V^\star$ is the apparent built-in potential: dashed lines, hole injection from MoO₃; solid lines, poly-HIL. MoO₃ was evaporated at 2 × 10⁻⁶ mbar.

This improved stability cannot be simply attributed to enhanced hydrophobicity of the $R_F$SIS anions. Although less than sulfonate, these anions are still rather hygroscopic. The degree of hydration is ca. 2.5–4 $H_2O$ per anion in the ambient[13], which is sufficient to fully de-dope the entire hole density of the polymer semiconductor several times over. Thus, the primary reason for improved stability is the downshifting of the effective donor level of their chemical DOS, as found in theory.

The $R_F$SIS anions also impart better thermal stability to the UHWF state. Although DL of the mTFF-SO₃ film decays to below 0.4 $h^+$/r.u. after a 5 min bake at 120 °C, the mTFF-C₂F₅SIS film survives above this level even after baking to 200 °C. This opens a sufficient window for thermal processing. Even the pTFF-C₂F₅SIS film can tolerate bake temperatures of up to 180 °C. This is superior even to the conventional SbF₆⁻ doped film. DL of mTFF: SbF₆ ($\phi$, 5.5 eV) decays to zero by 200 °C, due to fragmentation of SbF₆⁻.

**Tethered RFSIS: Ohmic hole contacts.** We confirm that these more stable UHWF films are indeed capable of Ohmic hole injection into semiconductors with deep ionization energies $I_E$. We use poly(9,9-bis(4-octylphenyl)fluorene-2,7-diyl) as benchmark semiconductor (PFOP, $I_E$, 5.8 eV) and glass/ITO/poly-HIL/ 110 nm PFOP/MoO₃/Ag as test structures[5], where poly-HIL is polymer hole injection layer, either hole-doped mTFF-C₂F₅SIS or pTFF-C₂F₅SIS, spin-cast in the glovebox or in ambient air. MoO₃ was evaporated and capped by Ag to provide a reference quasi-Ohmic top hole contact[3,28,29]. The conditions for Ohmic contacts in organic semiconductor devices have been specified in relation to accumulation carrier density and contact resistance[30]. We found the injected hole current density $J$ from poly-HIL is similar to that from MoO₃, independent of whether the poly-HIL is deposited in a glovebox or ambient air (Fig. 5). The built-in potential $V_{bi}$, measured by electroabsorption spectroscopy[31,32], is 0.0 ± 0.1 V, as expected. Thus, hole injection from these poly-HILs is Ohmic. Recently, pTFF-C₂F₅SIS has also been shown to provide good HIL for perovskite-based light-emitting diodes[33].

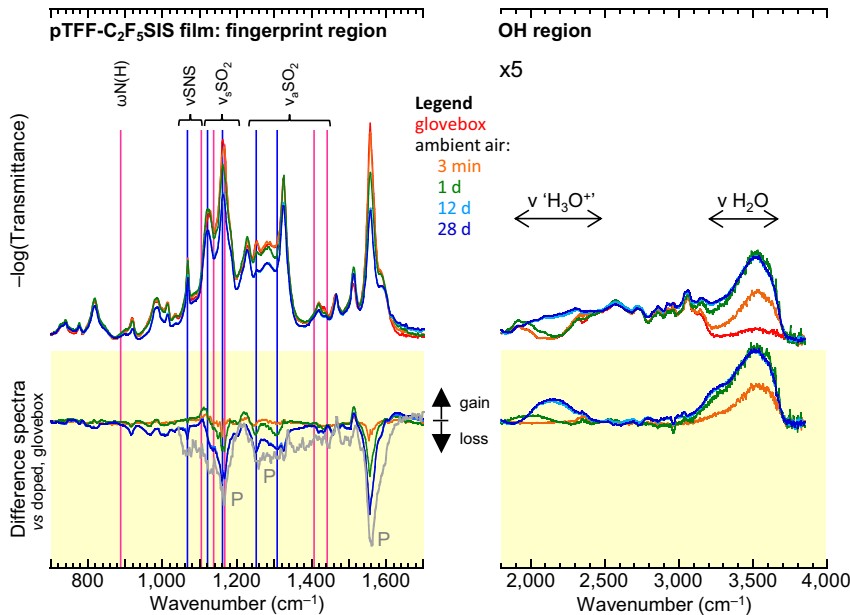

**Fig. 6 Evidence for ubiquitous water-oxidation reaction in $R_F$SIS-compensated polymer films.** Transmission FTIR spectra of pTFF-$C_2F_5$SIS film as a function of exposure time in ambient air (RH 65%, 295 K), together with difference spectra (yellow background). Blue lines mark characteristic vibrational modes of anion (–CH$_2$SO$_2$N$^-$SO$_2$C$_2$F$_5$); magenta lines, protonated anion (–CH$_2$SO$_2$N(H)SO$_2$C$_2$F$_5$). Grey spectrum is inverted polaron IRAV spectrum as a guide to the eye. The polaron absorption beyond 1800 cm$^{-1}$ is subtracted for clarity. Film thickness, 8 μm.

For 110-nm-thick PFOP films, the Mott–Gurney index given by: $m = \frac{d \log J}{d \log(V - V^*)}$, where $V^*$ is the apparent built-in potential, is 3.0, consistent with space-charge limited current transport in relatively thick disordered semiconductor films[34–36]. In contrast, PFOP films thinner than ca. 70 nm give the ideal $m$ value of 2.0 for $J$ larger than 300 mA cm$^{-2}$ (Supplementary Fig. 12a, b). Thin films can readily surpass the trap-filled limit. This yields a trap-free hole mobility of $3–4.5 \times 10^{-5}$ cm$^2$ V$^{-1}$ s$^{-1}$. With Al as hole-exit contact, the poly-HILs can still yield the same ideal $m$ index, but only at a higher $J$ due to lack of trap-filling at the exit contact (Supplementary Fig. 12c).

To illustrate their robustness, we employ variable-hole-doped pTFF-$C_2F_5$SIS films as injection models to investigate the threshold in $\phi$ for Ohmic injection. The DL of the hole injection film of pTFF-$C_2F_5$SIS is sequentially varied from 0.0 to 0.8 $h^+$/r.u. in 110-nm-thick PFOP diodes by controlled doping. Regardless of whether MoO$_3$/Ag, or Al, is employed as the hole-exit contact, we find that the hole current density saturates when DL exceeds ca. 0.4 $h^+$/r.u. (Supplementary Figs. 13 and 14, respectively). This indicates the $\phi$ threshold is 5.65 eV, providing further experimental validation that $\phi$ needs to approach within 0.15 eV of the conventionally defined $I_E$ for Ohmic injection to be realized. This agrees with earlier measurements using different hole injectors and test semiconductors[19,31]. However, tunnelling barriers, if present, would block Ohmic injection, as exemplified by fluoro-ionomer-modified poly(3,4-ethylenedioxythiophene):poly(styre-nesulfonic acid) films[37].

**Tethered $R_F$SIS: WOR is still limiting.** The chemical hole traps responsible for the residual de-doping of $R_F$SIS-compensated UHWF states, in ambient or when heated, are still the hydrated anions, demonstrating ubiquity of the WOR. Figure 6 shows de-doping is accompanied by generation of both '$H_3O^+$' (1100–1250 and 1900–2400 cm$^{-1}$) and protonated anion (conversion of –CH$_2$SO$_2$N$^-$SO$_2$C$_2$F$_5$ to –CH$_2$SO$_2$N(H)SO$_2$C$_2$F$_5$) on the time scale of de-doping. This points to a combination of Eqs. (4a) and

(4b). Although sorption of moisture completes within 10 min, as evidenced by $\nu$ H$_2$O band saturation, de-doping occurs with a characteristic time of 15–20 h, as evidenced by saturation of the IRAV bleaching. Spectral assignments have been authenticated (Supplementary Fig. 15). On the other hand, thermal de-doping in the nitrogen glovebox generates the protonated anion directly, similar to the situation with sulfonate (Supplementary Fig. 16). The extent of de-doping decreases with decreasing spectator cation size: NEt$_4^+$ > NMe$_4^+$ > Cs$^+$ ≈ Na$^+$, consistent with smaller cations lowering the donor level of the hole trap in the ion cluster.

**Self-protection against bulk water.** Finally, the $R_F$SIS-compensated UHWF states also exhibit an unexpected resilience to de-doping by liquid water, whose electrode potential corresponds to 5.25 eV. A hole-doped mTFF-$C_2F_5$SIS film was immersed in air-saturated water for 1 h in the ambient at 295 K and then dried in nitrogen for 1 h. Ultraviolet photoemission spectroscopy (UPS) showed its final $\phi$ is 5.65 eV, which is surprisingly almost the same as before. Its valence-band spectrum is also practically identical with the original (Fig. 7a)[19]. Contact with water should have de-doped the film to $\phi$ of about 5.25 eV. Indeed, a mTFF:SbF$_6$ test film exhibits $\phi$ of 5.1–5.2 eV after contact with liquid water (Fig. 7b). To probe the anomaly, we kept the mTFF-$C_2F_5$SIS film in ambient air, where hydration is substantially retained. The valence-band spectrum of this film indicates signs of de-doping[19] and $\phi$ drops to 5.4 eV. In addition, the key features of the valence band exhibits a rigid shift to lower binding energy that completely accounts for the $\phi$ reduction. This is the characteristic signature of a surface dipole. Thus, the hole-doped mTFF-$C_2F_5$SIS film, but not mTFF:SbF$_6$, is protected by spontaneous formation of a surface dipole (external polarity positive) that counteracts its inherent UHWF and blocks further hole injection into water. This dipole is reversible, so the mechanism confers a 'self-protection' to the UHWF state against liquid water. We speculate this dipole is driven by hydration of the spectator cations and their migration towards the water

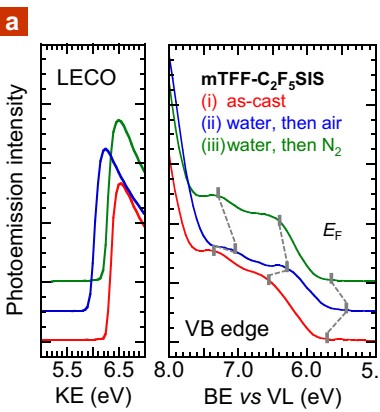
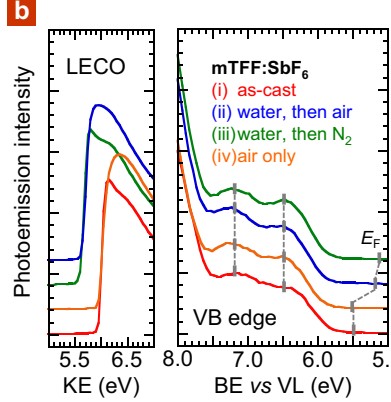

**Fig. 7 Self-protection against liquid water.** Normalized ultraviolet photoemission spectra of: **a** hole-doped mTFF-$C_2F_5$SIS film and **b** mTFF:$SbF_6$ film, (i) as-cast, after 1 h immersion in air-saturated water at 295 K, and then 1 h equilibration in (ii) ambient air, or (iii) nitrogen, before ultraviolet photoemission spectroscopy. A separate film (iv) after 2 h equilibration in ambient air, without contact with water, is also included in **b**. The low-energy cut-off (LECO) region is plotted against photoemission kinetic energy, whereas the valence-band edge (VB) region is plotted against binding energy from vacuum level. Spectra are shown offset for clarity. The Fermi energy ($E_F$) is marked, together with valence-band features from second-order derivatization.

interface, as conjugated polyelectrolytes are known to exhibit ionic layering of their tethered and free ions at the film surface[9]. Then, slowly drying the film in the glovebox relaxes this double layer, promoting recovery of the original $\phi$.

In summary, the water associated with the anion generally limits the stability of the UHWF state of hole-doped polymers exposed to ambient air and even those kept in the glovebox in two extreme anion systems tested, sulfonate and $R_F$SIS. This attests to the ubiquity of the WOR. Yet, it is possible to push its oxidative limit and, hence, workfunction limit of the doped polymer well beyond the 5.25 eV limit imposed by the FL of air-saturated liquid water. We demonstrate work functions up to 5.9 eV with good ambient and thermal stability. This strategy is to use super-acid anions with deep oxidation potential, together with small spectator cations, to destabilize hydronium generation from the water inevitably bound to the ion clusters, which can be further enhanced by blocking their slow relaxation. These insights would be relevant also to the design of soft materials for battery, bio-electronic and thermoelectric applications.

## Methods
**Computational methodologies**. DFT calculations were performed using the Gaussian 09 programme. Geometries were optimized and energies computed at the DFT/CAM-B3LYP/6-31++G($d,p$) level. We carried out several validation checks. The computed gas-phase $(H_2O)_2$ geometry and dissociation energy are given in Supplementary Table 1, and are compared with higher levels of theory and experimental results. The computed vertical and adiabatic ionization energies of gas-phase $(H_2O)_p$ clusters are given in Supplementary Table 2, and are compared with experimental ion appearance energies. The results indicate the quality of calculations is sufficient for our purpose. The computed gas-phase adiabatic electron-detachment energies $\Delta G_{D_g}$ for selected P+ X−$(H_2O)_p$ complexes are compiled in Supplementary Table 3, where X− ∈ {$PF_6^-$, $(CF_3SO_2)_2N^-$, $(C_2F_5SO_2)_2N^-$, $(CH_3SO_2)(CF_3SO_2)N^-$, $(CH_3SO_2)(C_2F_5SO_2)N^-$, $CF_3SO_3^-$, $CH_3SO_3^-$, Me(MeO)$PO_2^-$, $CF_3CO_2^-$, $CH_3CO_2^-$}, and P+ is simulated by tetramethylammonium (TMA+) as proxy for the hole polaron on the polymer semiconductor backbone. The geometries of both the initial TMA+ X−$(H_2O)_p$ and final TMA+ [X $(H_2O)_p$]● states are individually optimized to allow full relaxation of water in the vicinity of the anion. The correction of $\Delta G_{D_s}$ for differential Coulomb stabilization by the local ion cluster $\Delta(\Delta G_{Coul})$ was obtained from semi-empirical PM3 calculations and is compiled in Supplementary Table 4. The correction for differential matrix polarization and mobile-carrier screening was obtained from a polarizable continuum model, $\Delta(\Delta G_{pol}) = \Delta G_{pol,f} - \Delta G_{pol,i}$, using the generalized Born approximation, and is compiled in Supplementary Note 2.

**Materials**. A family of fluorene–alt–triarylamine polymers with tethered $SO_3$, $CF_3$SIS and $C_2F_5$SIS anions, and different spectator cation, e.g., Na+, Cs+, $NMe_4^+$ and $NEt_4^+$, were synthesized in-house following previously reported procedures[5,8,9,33]. $NOSbF_6$ and tris-($p$-bromophenyl)aminium

hexachloroantimonate (N($p$-Br$C_6H_4$)$_3$SbCl$_6$) were purchased from Sigma-Aldrich and were used as-received.

**Film-state doping**. In a typical procedure, a solution of mTFF-$SO_3$ in methanol was spin-cast in air onto desired substrates and baked on a hotplate at 220 °C for 15 min in a nitrogen glovebox. The film was contacted with 1 mM $NOSbF_6$ in anhydrous acetonitrile in the glovebox and spun off to dope the film. It was then contacted with anhydrous acetonitrile and spun off, twice, to remove excess dopant and salt by-product to give the self-compensated, hole-doped film.

**Solution-state doping**. In a typical procedure, the polymer solid was baked at 120 °C for 1 h in a nitrogen glovebox and then dissolved in anhydrous acetonitrile. Unless otherwise stated, 0.9 equivalent of oxidant, e.g., $NOSbF_6$ and N($p$-Br$C_6H_4$)$_3$SbCl$_6$, dissolved in anhydrous acetonitrile was added to hole-dope the polymer solution in the nitrogen glovebox. The doped polymer was then pre-cipitated with dimethyl carbonate to eliminate the soluble salt by-product. The precipitate was dried in vacuum oven overnight and then re-dissolved in anhydrous acetonitrile to give a solution of the self-compensated, hole-doped polymer. All operations were conducted in the nitrogen glovebox. This method is applicable to polymers with tethered $CF_3$SIS or $C_2F_5$SIS anions. Self-compensated, hole-doped polymer films can then be spin-cast from these solutions onto desired substrates. mTFF:$SbF_6$ was prepared by doping the mTFF solution in anhydrous acetonitrile with 0.9 equivalent of $NOSbF_6$, following the procedure in ref. [19].

**General spectroscopies**. Ultraviolet-visible near-infrared (UV-Vis-NIR) spectra were collected on an Ocean Optics QE Pro spectrometer in the nitrogen glovebox. Films were typically prepared on $O_2$-plasma-cleaned fused silica substrates. Fourier-transform infrared (FTIR) spectra were collected on a nitrogen-purged Nicolet 8700 spectrometer. Films were typically prepared on $O_2$-plasma-cleaned intrinsic silicon substrates and were mounted in a sample vacuum chamber fitted with KBr windows. The spectra of gas-phase molecular $H_2O$ and $CO_2$ were removed by subtraction, and an empirical universal function was used to remove curvature in the background.

**DL evaluation**. The UV-Vis-NIR spectrum of the film was collected in the glovebox at room temperature. DL was evaluated from both the loss in the neutral π–π* absorption intensity and the gain in the polaron $P_2$ intensity, with estimated uncertainty of ±0.05 $h^+$/r.u. The methodology was previously validated by XPS quantification[19].

**Doping stability test**. For ambient stability, 50-nm-thick films were kept in the ambient (RH 65%, 295 K) in the dark. For thermal stability, 50-nm-thick films were sequentially baked on a hotplate at 80 °C, 120 °C, 150 °C, 180 °C and 200 °C for 5 min each in the glovebox. For nitrogen stability, 50-nm-thick films were kept in the glovebox. Their UV-Vis-NIR spectra were collected in the glovebox at room temperature at selected time, or after each baking step.

**Vibrational spectroscopy of hole-doped mTFF-$SO_3$ film in wet $N_2$**. The film was mounted in a sample vacuum chamber fitted with KBr windows and kept in an atmospheric bag. Wet nitrogen made by passing through a saturated sodium nitrite solution was flowed into chamber. The chamber was placed in the FTIR spectro-meter after selected time intervals to collect the FTIR spectra.

**Ultraviolet and X-ray photoemission spectroscopies**. UPS and XPS were performed in this sequence in an ESCALAB UHV chamber equipped with an Omicron EA 125 U7 hemispherical electron energy analyser at a base pressure of <$10^{-9}$ mbar. In a typical procedure, 20 nm-thick films were spin-cast on $O_2$-plasma-cleaned Au-coated Si substrates in the glovebox, hermetically sealed in a nitrogen bag, and loaded into the UHV chamber load lock without exposure to the ambient. UPS was excited using He I radiation (21.22 eV). Photoemission normal to the film surface was collected. A sample bias of 5.00 V was applied to the sample to avoid cut-off artefact. The pass energy was 5 eV to give a resolution of 50 meV. Surface depletion and band-bending effects are not present in these heavily doped polymer films, whose work function is independent of film thickness from 5 to 50 nm. XPS was excited using MgKα X-rays (1253.6 eV). Photoemission normal to the film surface was collected. The pass energy was 20 eV to give a resolution of 0.7 eV.

**Device**. Current density–voltage (JV) characteristics were collected on a probe station in the glovebox using a Keithley 4200 semiconductor parameter analyser. Self-compensated, hole-doped mTFF-$C_2F_5$SIS or pTFF-$C_2F_5$SIS films were spin-cast from anhydrous acetonitrile solutions onto $O_2$-plasma-cleaned ITO–glass substrates to give 25 nm-thick films. PFOP films were spin-cast from toluene solutions. Al, $MoO_3$ and/or Ag were thermally evaporated through a shadow mask at a base pressure of $10^{-6}$ mbar to define the diodes.

## Data availability
Source data for all figures are available from the corresponding author.

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

## Acknowledgements
We thank Desmond W.Y. Teo and Yu Wang for contributions to the experimental work, and Sumitomo Chemical Co (SCC) for support. This research is partially funded by the National Research Foundation, Prime Minister's Office, Singapore, under its Competitive Research Programme (CRP Award Number NRF-CRP 11-2012-03: R-144-000-339-281, R-143-000-608-281).

## Author contributions
C.G.T. performed the calculations. Q.M.K. and C.G.T. performed the device fabrication and characterization, FTIR and UV-vis experiments. Q.M.K. performed the XPS and UPS. Q.M.K., M.C.Y.A., K.K.C. and Q.J.S. performed the syntheses. P.K.H.H. developed the theory. L.L.C. led the materials research programme. R.Q.P. led the materials heterostructure programme. All authors discussed the experiments and results. R.Q.P., L.L.C. and P.K.H.H. wrote the manuscript.

## Competing interests
The authors declare no competing interests.

**Additional information**

