## [Peer Review File · Nature Communications]

Reviewer #1 (Remarks to the Author):

The manuscript by Koh et al. describes a method to obtain high-workfunction doped polymer films with superior ambient and thermal stability. This is achieved by choosing large non-nucleophilic counter-anions, eventually preventing hole trapping by water complexes, which are inevitably present under ambient conditions. The design rules for achieving stable doped polymer layers are theoretically founded and experiments confirm that stable, high work function polymer films are achieved, which, importantly, can be processed in air.

I enjoyed reading the manuscript and I believe the work is very relevant, both in advancing understanding of hole-doped polymer layers, as well as the associated technological advances made. Although high work function materials are already available, air stability in combination with solution processability is currently lacking. I therefore believe that this manuscript would be suitable for publication in Nature Communications.

I have one comment that the authors should address:

Fig. 5 shows convincingly that hole injection from the hole-doped polymer is equally efficient as hole injection from the MoO₃/Ag electrode, which, on itself, is a respectable achievement for an air-processed hole-injection layer. However, it is not convincingly demonstrated that hole injection from these electrodes is Ohmic. I would like to encourage the authors to investigate a wider PFOP layer thickness range, and report the thickness dependence of the injected current density, which may reveal if an injection barrier is still present.

Reviewer #3 (Remarks to the Author):

Report on NCOMMS-20-08605 "Achieving ambient solution processable, ..." by Koh et al.

In this manuscript, the authors report their findings of stable ultrahigh-workfunction states beyond the bulk-water limit with thermal and ambient stability. DFT calculations were performed and doped thin films were characterized with UV-vis, FTIR, UPS and XPS. The Injection efficacy as well as temporal and thermal stability were also examined. Controlling the workfunction in ambient conditions is important for those in electronics and energy transformation and storage. The manuscript may be suitable for publication in Nat. Comm. after the following modifications/clarifications:

1. In the manuscript there are extensive discussions about the hydration of anions with water. The experimental verification, however, is with the dopant C₂F₅SO₃Na, a perfluoroalkylsulfonylimidosulfonyl derivative immiscible with water. Can there be a simpler explanation to the observed high workfunction and stability as simply due to the hydrophobicity of the dopant that keeps water away from the surface?
2. In Theoretical considerations, the first water molecule is taken to be bonded to the ion cluster and the subsequent ones hydrogen bonded. Is the nature of the first bonding ionic or of some other type?
3. In Computational methodology, separability and additivity of the dominant interactions are assumed. The authors should justify the assumption.
4. The data presented in Fig. 2 indicate that for weak acids the donor energy decreases as the cluster

size increases, while the reversed trend is observed for stronger acids. The authors should clarify if it is expected and what is the consequence.

5. Although UPS provides accurate measurements of the workfunction, it should be cautioned that complications may occur that affect the interpretation of the result. One possible complication is the photovoltaic effect if there's any surface band bending. Other factors include charging and dehydration in vacuum. Such possibilities in the present case should be discussed. Kelvin probe measurements may by-pass these problems.

6. Comparison of the measurements and calculations. While C2F5SIS is used in most of the experiments, there's no DFT calculation to compare with. It's peculiar that the authors did quite extensive calculations for potential dopants other than the one they actually used for the testing. If there's a reason behind it, they should explain it. In any case, the relevance of the calculation and the implications to the measured changes of the workfunction presented in Fig. 7 should be addressed.

7. The XPS in Fig. S5 is for mTFF-SO₃. The authors should provide their rationale on doing so instead of on the material systems that they studied to demonstrate the high workfunction. It should also be specified if the binding energy is defined from the vacuum level as in Fig. 7.

8. The doping level is deduced from UV-Vis measurements. The authors should check if such determination is consistent with the concentration analysis of the XPS core levels. More detailed analysis may even provide information on the hydration by detailed decomposition of the core levels, for example the O 1s.

9. The bias for the UPS measurements and pressure of the chamber should be provided.

10. The abbreviation pTFF is undefined. The molecular structures of the materials measured should also be provided in the Supplementary materials.

Detailed Response to Referee Reports

Reviewer 1

We thank this Reviewer for positive comments. The Reviewer has raised an important query, which we have now addressed with more experiments and a detailed note below.

1. "... it is not convincingly demonstrated that hole injection from these electrodes is Ohmic. I would like to encourage the authors to investigate a wider PFOP layer thickness range, and report the thickness dependence of the injected current density, which may reveal if an injection barrier is still present."

Response:

We have studied this problem in some detail over recent few years. The existence of traps complicates simple analysis, but the conclusion of Ohmic contact attained is robust. See the new data included in new Supplementary Fig S12 now. For thin films (< 100 nm) of the benchmark semiconductor PFOP (ionization energy, 5.8 eV) with mTFF-C₂F₅SIS as hole injection layer, and Al as hole-exit contact, the current density shows the expected V^2 scaling for current densities larger than about 500 mA cm⁻² that is characteristic of space-charge-limited conduction (SCLC) of the Mott–Gurney law (Fig S12c, old data from 24 months ago). For larger film thicknesses and/or lower current densities, the Mott–Gurney slope drifts above 2. This is characteristic of trap limited SCLC, which points to the existence of traps in polyfluorene, the greater the deviation, the deeper the traps, as is well known in the literature [Paul Blom and co-workers, *Nature Materials* 2019; Martijn Kemerink and co-workers, *Nature Materials* 2019]. To be very sure of this and address any remaining scepticism, we have made new experiments with the device configuration pTFF-C₂F₅SIS/PFOP/MoO₃, where the top evaporated MoO₃ acts as hole exit contact or hole injection contact, as desired, to provide the reference for the bottom polymer hole contact. The fitting of the PFOP film with two strong hole contacts at both its interfaces also better help to fill traps. We found indeed that the Mott–Gurney slopes for the thinner films and larger current densities (> 200 mA/cm²) are very close to 2, while the slopes for the thicker films and lower current densities also become uniformly closer to 2 (Fig S12a and b, new data from last month). The thickness dependence also scales as expected, around d^{-3} to d^{-4} . Most compellingly, the hole current injected from the spin-on polymer contact is actually (slightly) larger than that injected from the evaporated MoO₃ contact. Since MoO₃ is well known to be practically Ohmic, this observation together with that of the correct Mott–Gurney index allows us to be sure that the polymer hole contact is Ohmic.

Reviewer 3

We thank this Reviewer for attention to many details. We overlooked to report some of these in the original manuscript – sorry. These are now compiled in various new Supplementary Figures, including Figure S5 for the chemical structures of the materials. Our line-by-line responses are as follows.

1. “Can there be a simpler explanation to the observed high workfunction and stability as simply due to the hydrophobicity of the dopant that keeps water away from the surface?”

Response:

This is the natural starting assumption, and was indeed our first working hypothesis. Detailed thermogravimetry however shows that the perfluoroalkylsulfonylimidosulfonyl anions are still fairly hygroscopic, binding about half as many water molecules as sulfonate anions, although they no longer confer water solubility to the polymers. See Figure below. In the hole-doped state of mTFF-C₂F₅SIS-Na, moisture is present at the level of *ca.* 3 H₂O per anion in the ambient (22°C, 60% RH; Teo et al, ACS Appl. Mater. Interfaces 11 (2019) 48103). This amount is sufficient to ‘kill off’ all the holes even if one sixth of the water can act as chemical trap, since two anions are present for each hole, one of them counterbalanced by a spectator cation for solubility. FTIR measurements show the tethered perfluoroalkylsulfonylimidosulfonyl-anion films in fact sorb moisture very rapidly from the ambient, Supplementary Figure S10. This is also made clear by Figure 6, which shows water sorption on two time scales, but de-doping occurs on the slower timescale. The ‘hydrophobicity’ of the dopant anions is not sufficient to keep water away from the system.

In separate studies, we found that polyelectrolyte systems with both large ‘hydrophobic’ cations, like tetramethylammonium and tetraphenylphosphonium, and large ‘hydrophobic’ anions, like the ones here, are still hygroscopic, binding a significant amount of water in the as-deposited films. We traced this to the coulomb binding of water in molecular cavities in these materials.

Thus the reason why water can surprisingly be compatible with ultrahigh work-function states is due to energetics. The most vulnerable water associated with large anions can have their HOMO level sufficiently depressed to avoid becoming a chemical hole trap. Were this not the case, it would have been impossible to achieve air stability for any solution processable ultrahigh-workfunction polymer, since elimination of water is not really possible in ‘soft’ matter containing ions.

Figure caption: **Segmented TGA of water desorption from selected polymers in flowing nitrogen.** (a) mTFF-SO₃-Na. (b) mTFF-C₂F₅SIS-Na. Solid line, first scan; dotted line, second scan, after re-equilibrate in ambient (22°C, 70% RH; 1 h). Data starts below 100% mass because of desorption during setup before reaching first isothermal set point. The ratio of H₂O/anion is annotated for each step, based on the dry weight obtained at the end of segment (iii). Sample form: powder, precipitated from solution. Figure from cited ref 13.

2. “Is the nature of the first bonding ionic or of some other type?”

Response:

We assume the Reviewer is referring to the bonding between the various water species and the ion cluster. The bonding of water to the anion is by hydrogen-bonding. This is clear from the DFT calculations. [Yes, DFT/CAM-B3LYP/6-31++G(d,p) appears to reproduce H-bonding fairly well, based on benchmark calculations.] When the water is ionized, a hydronium ion forms upon geometric relaxation, as one would expect. The binding to the ion cluster comprises both coulomb and H-bonding. We have allowed geometric relaxation for the water molecules in the vicinity of the ion cluster. The results suggest the natural formation of H bonding network amongst the water molecules and also with the attached anion. This is now briefly discussed in the Method section.

3. “... separability and additivity of the dominant interactions are assumed. The authors should justify the assumption.”

Response:

This is a necessary assumption in order to factorize the gargantuan degrees of freedom in the configuration space to smaller subsets that can be handled by calculations, in the same spirit as the Born–Oppenheimer approximation. The essential idea is:

- (1) Quantum mechanical interactions, including H-bonding, are of very short range. Therefore the hydrated anion complex $X^-(H_2O)_p$ is treated as a quantum mechanical entity.
 - (2) These energies are then shifted by the 'background' Coulomb potential set up at $X^-(H_2O)_p$ by the other surrounding ions. The ion-cluster configurations are sampled by MM2/PM3 and their energies computed at the less costly PM3 level since these configurations are primarily determined by Coulomb interactions between the constituent ions subjected to steric and tethering constraints.
 - (3) Longer-range polarization of the matrix is evaluated by a classical polarizable continuum model.
4. "The data presented in Fig. 2 indicate that for weak acids the donor energy decreases as the cluster size increases, while the reversed trend is observed for stronger acids. The authors should clarify if it is expected and what is the consequence."

Response:

We thank the Reviewer for pointing out that we did not explain so well in the earlier manuscript draft. Sorry. We have now improved the discussion with one new paragraph in the main text. Basically the differences have a really intuitive explanation. For weak acid anions, the product of chemical hole trapping in the hydrated anion complex is a radical, not a hydronium. Calculations show the radical is primarily derived from the anion. In this case, the initial anion charge is stabilized by water molecules. So as the hydrating water cluster size increases, the hydrated anion donor level becomes deeper, i.e., more difficult to detach its electron. For strong acid anions, the product is a hydronium. In this case, the final hydronium charge is stabilized by hydration. So as the water cluster size increases, the anion...water donor level becomes shallower.

The dependence on the ion cluster size, i.e., number of ion pairs in the local cluster, is also in opposite directions for these two cases. For the weak acid anions, increasing ion cluster size increases stabilization of the initial anion charge (Madelung effect). This lowers the donor level. For the strong acid anions, increasing ion cluster size also increases stabilization of the final hydronium charge, but this raises the donor level, as the hydronium is now easier to form.

Since hydration effects level off beyond 5 H₂O per anion in the ambient, the donor energies in Fig 2 provide an estimate of the limit of stability of ultrahigh work functions for different hydration number and ion-cluster size. We have expanded three paragraphs to clarify these now.

5. "Although UPS provides accurate measurements of the workfunction, One possible complication is the photovoltaic effect if there's any surface band bending. Other factors include charging and dehydration in vacuum. Such possibilities in the present case should be discussed."

Response: Yes, we have included a simple discussion now. Below is our detailed response.

Surface band bending. The materials are in the strong doping regime, doping level $\geq 10\%$ of repeat units, corresponding to a few 10^{20} cm^{-3} . At such high carrier density, surface depletion and band bending effects are not present. We know this is the case also because the measured work function is independent of film thickness from 5 to 20 nm. So there is no significant surface photo voltage effect.

Charging. The doped films have an electrical conductivity of $\geq 10^{-3} \text{ S cm}^{-1}$, depending on semiconductor core. UPS and XPS were performed on films usually less than 20-nm thick. The photoemission current is typically 1 nA mm^{-2} . The voltage drop is negligible, much less than μV . We know this is the case also because there is no shift in XPS core level between the doped and undoped films.

Dehydration. In extensive work, we have established the work function is set by the electronic structure of the polymer and its doping level, modified by Madelung potential of the counter-ion and other spectator ions in the vicinity of the carrier. During UPS/XPS measurement, the film retains tightly-bound water at the level of 1–1.5 H_2O per anion. This water is not ionized nor oriented, and so does not significantly affect work function. We know this is the case because the effective work function measured in the diodes by electroabsorption spectroscopy (where more water is present) agrees with the UPS vacuum work function to within 0.1 eV [Ang et al, Mater Horizon (2020), doi 10.1039/c9mh01749f]. Furthermore, unlike band solids, whose work function can be significantly affected by sorbed water on the surface due to molecular orientation, the water if present in these materials is isotropically distributed in the bulk. We know this is the case because variable XPS together with UPS confirms the absence of any variable surface dipole larger than 0.1 eV [Ang et al, Mater Horizon (2020), doi 10.1039/c9mh01749f].

6. “While C₂F₅SIS is used in most of the experiments, there’s no DFT calculation to compare with. ... the relevance of the calculation and the implications to the measured changes of the workfunction presented in Fig. 7 should be addressed”

Response:

We have fixed this gap now. We have added one more model anion for C₂F₅SIS, (CH₃SO₂)(CF₃SO₂)N⁻ to Figure 2, Supplementary Table S3. The attachment of alkyl chain to the methyl end does not shift energetics much. We have also improved the discussions on anion effects. Please note that ions in Figure 2 are only model ions. They are not directly useful as counter-ions for hole-doped polymer organic semiconductors, because of ‘dopant migration’. To overcome this, we need to immobilize them, for example by tethering to alkyl chains. Tethering through the alkyl moiety does not change the energetics of the hydrated anion. Indeed, experimentally measured stability tracks the predicted stability limits remarkably well for a simple zero-free-parameter model. Re Figure 7, we have included one more experimental set now, in new Figure 7b. This

experimental set confirms that the measured recovery in work function in Figure 7 (now Figure 7a) is a feature of self-compensated, hole-doped polymers, due to the serendipitous formation of a reversible ‘protection’ surface dipole in the presence of liquid water. The presence of dipole is now confirmed by the rigid VB spectra shift observed in the self-compensated hole-doped polymer, but not in the one that is conventionally doped with SbF_6^- .

7. “The XPS in Fig. S5 is for mTFF-SO₃. The authors should provide their rationale on doing so instead of on the material systems that they studied to demonstrate the high workfunction. It should also be specified if the binding energy is defined from the vacuum level as in Fig. 7”

Response:

Yes, all binding energies are referenced to vacuum level. We have now made clear in the figure captions. The reason why we presented XPS of mTFF-SO₃ in Fig S5 is to point out the excessive loss of Na and conversion of SO_3^- to SO_3H in the widely used sulfonate systems that somehow has escaped the other investigators. Following Reviewer’s suggestion, we have now added a new XPS measurement on the more desirable system pTFF-C₂F₅SIS to show that this is indeed free of such problems, in new Supplementary Figure S9.

8. “The doping level is deduced from UV-Vis measurements. The authors should check if such determination is consistent with the concentration analysis of the XPS core levels. More detailed analysis may even provide information on the hydration by detailed decomposition of the core levels, for example the O 1s.”

Response:

For the triarylaminium polymer cores, hole doping leads to the emergence of P_1 and P_2 subgap polaron absorptions at ca. 1.0 and 2.5 eV respectively, similar to molecular triarylaminium salts. For these salts and polymers, it is well known that the polaron absorption band intensity scales linearly with the doping level, as also is the loss of the $\pi^* \leftarrow \pi$ absorption band intensity. Therefore we used the fractional bleaching of the $\pi^* \leftarrow \pi$ absorption band to estimate the hole doping level, calibrated using the undoped spectrum (0 h^+ / per repeat unit) and fully-doped spectrum (1 h^+ / per repeat unit).

Such an evaluation has been firmly established by XPS quantification for a model system through ratio of core intensity of N_{1s}^+ to total N_{1s} , and SbF_6^- to total N_{1s} [Png et al, Nat Commun 7:11948 (2016)]. The method however cannot be applied to R_FSIS-compensated films, because R_FSIS N_{1s} overlaps with triarylamine N_{1s} in the undoped state, and with the triarylaminium N_{1s} in the doped state.

We have studied the O1s intensities now to infer that H₂O is present at 1.0–1.5 H₂O per anion. We have mentioned this in the text now.

9. “The bias for the UPS measurements and pressure of the chamber should be provided.”

Response:

Bias is -5.00 V. Chamber pressure is $< 10^{-9}$ mbar. We have included these in the Experimental now.

10. “The abbreviation pTFF is undefined. The molecular structures of the materials measured should also be provided in the Supplementary materials”

Response:

We have rectified this now. The chemical structures can be found in new Supplementary Figure S5.

Reviewer #1 (Remarks to the Author):

After reading the reviewer comments and the corresponding response of the authors, I believe that the manuscript is now in a sufficiently complete state and of sufficient quality to be published in Nature Communications.

Reviewer #3 (Remarks to the Author):

The authors have substantially improved the manuscript. The responses are extensive, but in a few places geared toward convincing the reviewer instead making changes for the readers. Specifically, I'd like to see the explanations to Reviewer 3's Point 1 & 3 be propagated into the text or Supplementary Materials.